

# Moderately low nitrogen application mitigate the negative effects of salt stress on annual ryegrass seedlings

An Shao[*], Zhichao Sun[*], Shugao Fan, Xiao Xu, Wei Wang, Erick Amombo, Yanling Yin, Xiaoning Li, Guangyang Wang, Hongli Wang and Jinmin Fu

Coastal Salinity Tolerant Grass Engineering and Technology Research Center, Ludong University, Yantai, China

[*] These authors contributed equally to this work.

Corresponding author
Jinmin Fu, turfcn@qq.com

## ABSTRACT

Appropriate application of nitrogen (N) can alleviate the salt stress-induced damage on plants. This study explores the changes of nitrogen requirement in feeding annual ryegrass seedlings under mild salt concentrations (50 mM, 100 mM) plus its underlying mitigation mechanism. Results showed that low salt concentration decreased N requirement as observed from the increment in plant height and biomass at a relative low N level (2.0 mM not 5.0 mM). Under salt treatment, especially at 50 mM NaCl, the OJIP (Chl a fluorescence induction transient) curve and a series of performance indexes ($PI_{ABS}$, $RC/CS_0$, $ET_0/CS_0$, $\phi E_0$, $\phi_0$) peaked whereas $DI_0/RC$, $V_j$ and $M_0$ were the lowest under moderately low N level (2.0 mM). In addition, under salt stress, moderately low N application could maintain the expression of NR (nitrate reductase) and GS (glutamine synthetase) encoding genes at a relatively stable level but had no effect on the expression of detected NRT (nitrate transporter) gene. The seedlings cultured at 2.0 mM N also exhibited the highest activity of CAT and POD antioxidant enzymes and the lowest MDA content and EL under relative low level of salt treatment. These results indicated that mild salt treatment of annual ryegrass seedlings might reduce N requirement while moderately low N application could promote their growth via regulating photosynthesis, alleviating ROS-induced (reactive oxygen species) damage and maintenance of N metabolism. These results also can provide useful reference for nitrogen application in moderation rather than in excess on annual ryegrass in mild or medium salinity areas through understanding the underlying response mechanisms.

## INTRODUCTION

Feeding annual ryegrass (*Lolium multiflorum* Lam.) is an important forage grass with high yield, good palatability and high nutritive value (*Castanheira et al., 2014*). Salt stress is one of the major abiotic factors that limit annual ryegrass growth and productivity. The adverse effects of salt stress on plants include ionic toxicity, osmotic stress and secondary stresses, such as a decline in photosynthesis, oxidative stress and nutritional disorders (*Allakhverdiev & Murata, 2004*; *Kalaji et al., 2011*; *Zhu, 2001*). The forage quality such as crude protein,
organic matter can be adversely affected by elevated salinity (*Ibrahim et al., 2018*; *Robinson et al., 2004*). In response, through long term evolution, plants have adopted a series of response mechanisms to resist and minimize salt stress-induced damage (*Deinlein et al., 2014*; *Zhao et al., 2020*). For example, in response to salt-induced oxidative stress, a series of antioxidant enzymes were induced to scavenge excessively produced reactive oxygen species (ROS) (*Kohler et al., 2009*; *Baby & Jini, 2010*), such as superoxide dismutase (SOD), peroxidase (POD), and catalase (CAT) (*Apel & Hirt, 2004*; *Dong, Kim & Lee, 2001*).

In saline environment, nitrogen (N) is usually the limiting growth macro-nutrient, thus N application is the most effective method of improving plant growth under salt stress. The mainly available inorganic N forms are ammonium N ($NH_4^+$) and nitrate N ($NO_3^-$), which are transported by AMT (Ammonium transporter) and NRT (Nitrate transporter) respectively (*Giagnoni et al., 2015*). The inorganic N is then assimilated and converted into amino acid via several enzymes such as nitrate reductase (NR), glutamine synthetase (GS) and glutamine 2-oxoglutarate aminotransferase (GOGAT) (*Xu, Fan & Miller, 2012*). Salt stress has also proved to inhibit the activity of many enzymes such as NR and GS/GOGAT, involved in N assimilation in maize (*Zea mays*), cowpea (*Vigna unguiculata*) mung bean (*Vigna radiata*), tomato (*Solanum lycopersicum*) and rice (*Oryza sativa*) (*Chakrabarti & Mukherji, 2007*; *Debouba et al., 2007*; *Khan & Srivastava, 1998*; *Parul, Kumar & Kumar, 2015*; *Silveira et al., 2001*; *Wang et al., 2012*), and then decreased the absorption and utilization of N in plants (*Singh, Singh & Prasad, 2016*). Increased salinity was also reported to significantly reduce nitrogen use efficiency (*Murtaza et al., 2013*). On the contrary, some studies showed that processes related to N uptake and assimilation in some plant species were stimulated under certain levels of salt stress. For instance, salt could induce the expression level of nitrate transporter genes such as *McNRT1* (*Popova, Dietz & Golldack, 2003*). Under salt exposure, the nitrate uptake rate and activity of NR were promoted in glasswort (*Salicornia europaea*) (*Nie et al., 2015*). Thus, it has been speculated that the alteration of plant N nutrition level may hold great promise for regulating salinity response in different species under certain salt level (*Chen et al., 2010*).

Generally, N application can ameliorate the negative effects of salinity by compensating and correcting nutritional imbalances in higher plants (*Esmaili et al., 2008*; *Gómez et al., 1996*; *Mansour, 2000*; *Villa et al., 2003*). Several N-containing compounds were accumulated in plants that were subjected to salt stress (*Dluzniewska et al., 2007*; *Ehlting et al., 2007*; *Sudmalis et al., 2018*). The accumulates were then reported to participate in osmotic adjustment, promotion of photosynthetic capacity and mitigation of oxidative stress by scavenging excessive ROS (*Homaee, Feddes & Dirksen, 2002*; *Mansour, 2000*; *Rontein, Basset & Hanson, 2002*; *Song et al., 2006*; *Todorova et al., 2013*). Although many studies have shown that N plays an important role in the amelioration of salt stress, it has been confirmed that the alleviation of salt inhibition from N application shows concentration effects. For example, at the higher salinity levels, increasing the N content was found to be ineffective in resisting negative influences caused by the enhanced salt concentrations in tomato (*Papadopoulos & Rendig, 1983*). Another previous study reported that low levels of N could mitigate the negative effects whereas high N levels may exacerbate the detrimental effects of salt stress on photosynthetic rate of chickpea leaves (*Soussi,*

*Ocaña & Lluch, 1998*). A recent study also found that, low to moderate N application could mitigate the adverse effects, while excessive N could elevate the negative effects of salt stress on cotton growth (*Chen et al., 2010*). In maize, reduction in N application could not cause additional damage on plant development under salt stress based on reductions in evapotranspiration (*Lacerda et al., 2016*). Other studies also pointed out that excessive nitrogen fertilization under high salinity levels might lead to more pronounced osmotic effect which in turn provokes the negative effect on crop yield (*Beltrão et al., 2002*). In addition, in highly saline soils, excessive application of N fertilizer will trigger secondary soil salinization, which in turn adversely affects crop growth (*Chen et al., 2010*). Moreover, over fertilization with N may contribute to N leaching in the saline soil, where plants are unable to utilize the supplied N fertilizer efficiently and leading to groundwater pollution (*Pessarakli & Tucker, 1988*; *Shenker, Ben-Gal & Shani, 2003*; *Ward, 2013*).

Therefore, N requirements for plants under saline environment might be inconsistent with those in normal environment. The difference might be attributed to distinct physical and chemical properties of growth medium and the alteration of plants nitrogen use efficiency plus other physiological response. Hence, in order to minimize the negative effects of salinity as well pollution of underground water, proper N fertilizer management in plants is necessary for different salt conditions. A previous study on annual ryegrass reported that increasing N concentration in the nutrient solution enhanced shoot biomass production under relatively high salinity levels (*Sagi et al., 1997*). However, at relatively low salt concentrations, the optimization of nitrogen demand and the possible mechanisms underlying this alleviation in annual ryegrass are still not fully explored. Based on the above studies, the objectives of this work were to assess the optimal N level under relatively low salt level and to investigate the possible salt stress-alleviatory mechanisms by analyzing physiological indexes and metabolism of N in annual ryegrass seedlings.

## MATERIALS AND METHODS

### Plant materials and growth conditions

Annual ryegrass seeds were thawed in plastic pots that were filled with plant growth medium and then cultured in the greenhouse under natural sunlight. After one month, the seedlings were then transferred into Erlenmeyer flasks containing 585 mL of nutrient solution. The seedlings were then mowed to a height of 12.5 cm before the treatments were initiated. The experiment was set as control (0 mM NaCl) and NaCl treatment (50 mM or 100 mM). Both the control and NaCl regimes included different nitrogen application level (using $NH_4NO_3$ as nitrogen source). The hydroponic culture media was processed in a growth chamber under the following conditions: 22/18 °C (day/night), 65% relative humidity, 300 $\mu$mol m$^{-2}$s$^{-1}$ photons and a 16-h day/8-h night cycle. The culture solution was refreshed every two days.

### Chlorophyll *a* fluorescence transient and the JIP-Test

A pulse amplitude modulation fluorometer (PAM2500; Heinz Walz GmbH) was used to measure the chlorophyll a fluorescence transient. In brief, the annual ryegrass leaves were put in dark place for 30 min, and the leaves were then exposed to 3,000 $\mu$mol photons

$m^{-2}$ $s^{-1}$ red light condition. Each treatment consisted of at least three replicates. Based on the theory of energy fluxes in biofilm, the JIP test can further translate the primary data into other biophysical parameters (*Force, Critchley & Rensen, 2003*). Hence, the basic parameters were used to calculate a series of parameters (*Yusuf et al., 2010*).

### Chlorophyll content and electrolyte leakage

The SPAD 502 Plus Chlorophyll Meter (SPAD-502Plus; Spectrum Technologies, Inc., Dallas/Fort Worth, TX, USA) was used to quantify the leaf chlorophyll content. The electrolyte leakage (EL) was measured according to the previous method (*Blum & Ebercon, 1981*).

### Enzymes activity and lipid peroxidation

Fully expanded of 0.3 g leaves were immediately grounded into powder with liquid N. four mL ice-cold phosphate buffer (50 mM, pH 7.8) was added into the powder and the samples were centrifuged at 12,000 rpm for 20 min at 4 °C. The supernatant was collected to measure the activity of POD and CAT and the content of MDA based on the method as described by previous study (*Fu & Huang, 2001*).

### Quantitative RT-PCR Analysis

The total RNA was isolated and reverse transcribed using the RNeasy kit (Qiagen) and TaqMan reverse transcription kit (Applied Biosystems). Quantitative real-time RT-PCR analysis was conducted using SYBR Green real-time PCR master mix (Toyobo, Japan) and ABI real-time PCR system (Applied Biosystems, Foster City, CA, USA). The primers used are listed in Table S1. The ryegrass *Actin* gene was used as an inner control, and comparative Ct method was applied for gene expression level analysis.

### Statistical analysis

One-way ANOVA was performed using SPSS17.0 for Windows (SPSS). All of above tests had at least three independent replicates. Results were expressed as mean $\pm$ *SD*, and letters show significant differences $(P < 0.05)$ by Student's *t*–test.

## RESULTS

### Effect of different N level treatment on the growth of annual ryegrass seedlings under NaCl stress

Under control condition, the plant height and the relative increase of biomass of annual ryegrass seedlings peaked under 5.0 mM N, and then slightly declined under 10 mM N. As compared to 5.0 mM N, the height of seedlings grown under 2.0 mM N and 10 mM N declined by about 10% and 15% respectively (Figs. 1A–1D). However, under 50 mM or 100 mM NaCl treatment, the plant height reached maximum under 2.0 mM N and then declined slightly under 5.0 mM N. In addition, salt treatment dramatically reduced the height of plants grown under 5.0 mM or 10 mM N compared with their respective control plants. However, under 2.0 mM N condition, the plant height had no significant difference with or without salt treatment (Fig. 1D). When exposed to 50 mM NaCl, the relative increase of biomass had no significant difference among different N levels. After the

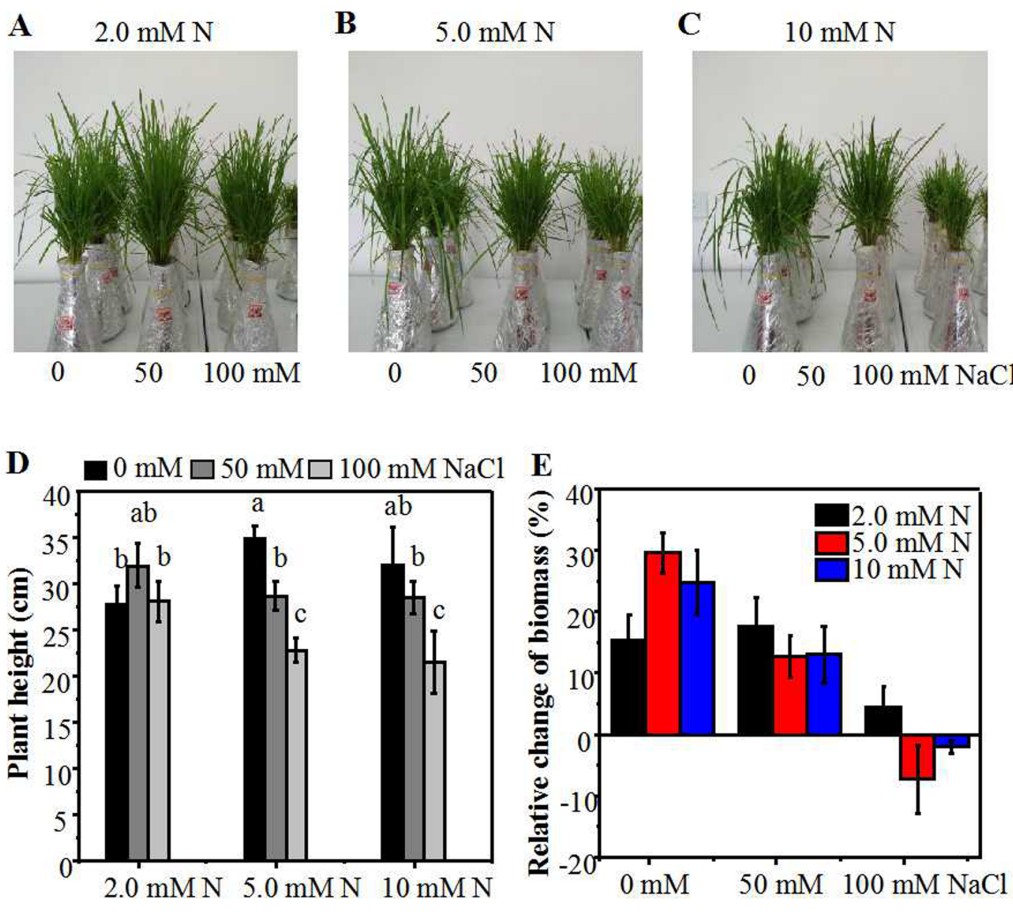

**Figure 1  Morphological parameters of annual ryegrass seedlings grown under different nitrogen and salt conditions.** Morphological parameters of annual ryegrass seedlings grown under different nitrogen and salt conditions. The seeds of annual ryegrass were cultured in soil for one month, and the seedlings cut to the same height were then transferred into different nitrogen level (2.0, 5.0, 10 mM) under NaCl (0, 50, 100 mM) stress in a hydroponic culture. After being grown for 10 days, the plant height and biomass were measured. (A–C) Images of seedlings at 10 days after transferred. (D) Plant height at 10 days after transferred. (E) The relative change of biomass (% of biomass which was measured before treatment). Different letters above the columns indicate significant differences at $P < 0.05$ by Student's $t$–test.

plants were exposed to 100 mM NaCl for 10 days, the biomass only increased when 2.0 mM N was applied whereas the biomass decreased when extra nitrogen (5.0 or 10 mM N) was applied (Fig. 1E). These similar alteration trends in biomass and plant height suggested that salt stress might change the N requirement of annual ryegrass and moderately reducing N application might alleviate the inhibitory effect of salt stress on annual ryegrass seedlings growth. In order to investigate whether ultra-low N treatment still had a moderating function on the growth of annual ryegrass seedlings under salt stress, we added a lower concentration (0.5 mM) of N treatment and conducted another experiment. The results showed that the plant height of annual ryegrass seedlings achieved maximum under 5.0 mM N without salt treatment. However, when exposed to 50 mM or 100 mM NaCl, the plant height reached maximum under 2.0 mM N (Figs. S1A, S1B), showing the similar

tendency with the first experiment (Figs. 1A, 1B). In addition, without NaCl treatment, there was no significant difference of plant height when seedlings were supplemented with lower N concentration (0.5 mM or 2.0 mM). However, under NaCl treatment, the plant height was significantly increased under 2.0 mM N compared with 0.5 mM N (Fig. S1). The results of these two experiments at different times together confirmed that the alleviatory effect of N application on the growth inhibition of annual ryegrass under salt stress might have a certain range. Moderately low N could alleviate the inhibition of annual ryegrass growth by salt stress through a series of response mechanisms, whereas ultra-low N could not promote, but seriously inhibit the growth of annual ryegrass.

## Impact of N on the OJIP transient curve in the leaves of annual ryegrass under NaCl stress

In order to decipher the effect of moderate N-mediated alleviation of salt stress on annual ryegrass, the impact of N levels on photochemistry of photosystem II (PS II) of NaCl treated annual ryegrass seedlings were determined through chlorophyll a fluorescence transient-JIP test. The step O to J represents the reduction process of $Q_A$ by PSII. Due to the brimming of the plastoquinone pool, the curve then rose to I phase. The step I to P was an account for the blockage of electron transfer to the acceptor side of PSI. Under control condition, the fluorescence of I and P phase of seedlings leaves grown with 2.0 mM N or 5.0 mM N was stronger than that grown with 0.5 mM N (Fig. 2A). However, after NaCl exposure, the chlorophyll fluorescence curve of annual ryegrass leaves grown with 2.0 mM N from I to P step was higher than that under 0.5 mM or 5.0 mM N (Figs. 2B, 2C). In particular, the OJIP curve was much higher when plants exposed to relatively low NaCl treatment (50 mM) under 2.0mM N level compared to other two N levels (Fig. 2C). The results suggested that deficient or excessive nitrogen application under salt stress might lead to the photosynthetic electron transport traffic jam, especially beyond $Q_A{}^-$. In addition, under NaCl treatment, the leave chlorophyll content of the plants grown with 2.0 mM and 5.0 mM N was significantly higher than that grown with 0.5 mM N. However, there was no significant difference in the chlorophyll content between 2.0 mM and 5.0 mM N-supplied plants (Fig. 2D).

## Impact of N on Chlorophyll fluorescence parameters in the leaves of ryegrass under NaCl stress

The chlorophyll fluorescence parameters were used to quantify the photosynthetic behavior of the samples. Under the control condition, the $PI_{ABS}$ value, which represents the overall activity of PSII, increased with the N level, and peaked under 5.0 mM N (Fig. 3A). However, under 50 mM or 100 mM NaCl treatment, the $PI_{ABS}$ value under 2.0 mM N were higher than that under other N levels (Figs. 3B, 3C). The variable fluorescence at J phase ($V_j$) and the relative speed of $Q_A$ deoxidation ($M_0$) of NaCl-treated leaves grown with 2.0 mM N were smaller than those grown with 0.5 mM or 5.0 mM N, and the difference was most significant under 50 mM NaCl treatment (Fig. 3B). Under normal condition, $\Psi_0$ and $\Phi E_0$ had no significant difference among three N levels (Fig. 3A). When exposed to 50 mM NaCl, the proportion of energy used for photochemical reaction and energy electron transport in leaves ($\phi_0$, $\phi E_0$) grown with 2.0 mM N were larger than those grown with other N levels,

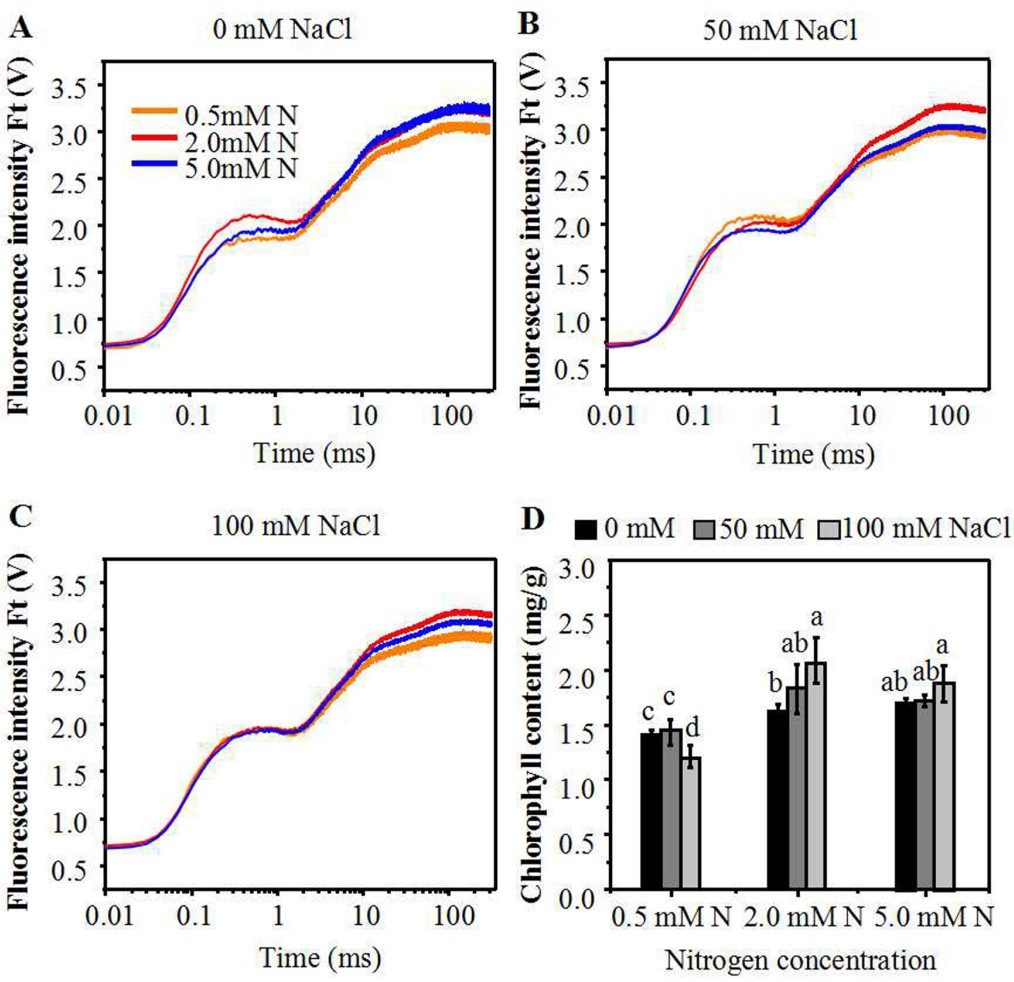

**Figure 2** **Alterations of chlorophyll fluorescence transients in leaves of annual ryegrass.** The annual ryegrass were grown with different nitrogen concentrations (0.5, 2.0, 5.0 mM) under 0 mM, 50 mM (B), 100 mM NaCl (C) stress respectively. (D) Influence of nitrogen concentration on chlorophyll content under different levels of NaCl stress respectively. Different letters above the columns indicate statistically significant differences at $P < 0.05$ by Student's $t$–test.

together with greater reaction center density $RC/CS_0$ and electron-transfer energy $ET_0/CS_0$ and lower $DI_0/CS_0$ (the energy consumed in unit cross-sectional area) (Fig. 3B). However, $\phi P_0$, which represents the maximum quantum yield for primary photochemistry, displayed no changes. When exposed to 100 mM NaCl, the $PI_{ABS}$ value under 2.0 mM N was higher than that under other N levels, whereas the other parameters showed no significant change (Fig. 3C). These results suggested that the optimum amount of N might promote primary photochemical reactions of PSII, especially under relatively low NaCl level.

## The lipid peroxidation levels and activities of antioxidant enzymes in the leaves of the annual ryegrass seedlings under NaCl stress

Malondialdehyde (MDA) is one of the products of membrane lipid peroxidation which can be used to assess the degree of the salt-induced damage to plants. The results showed that,

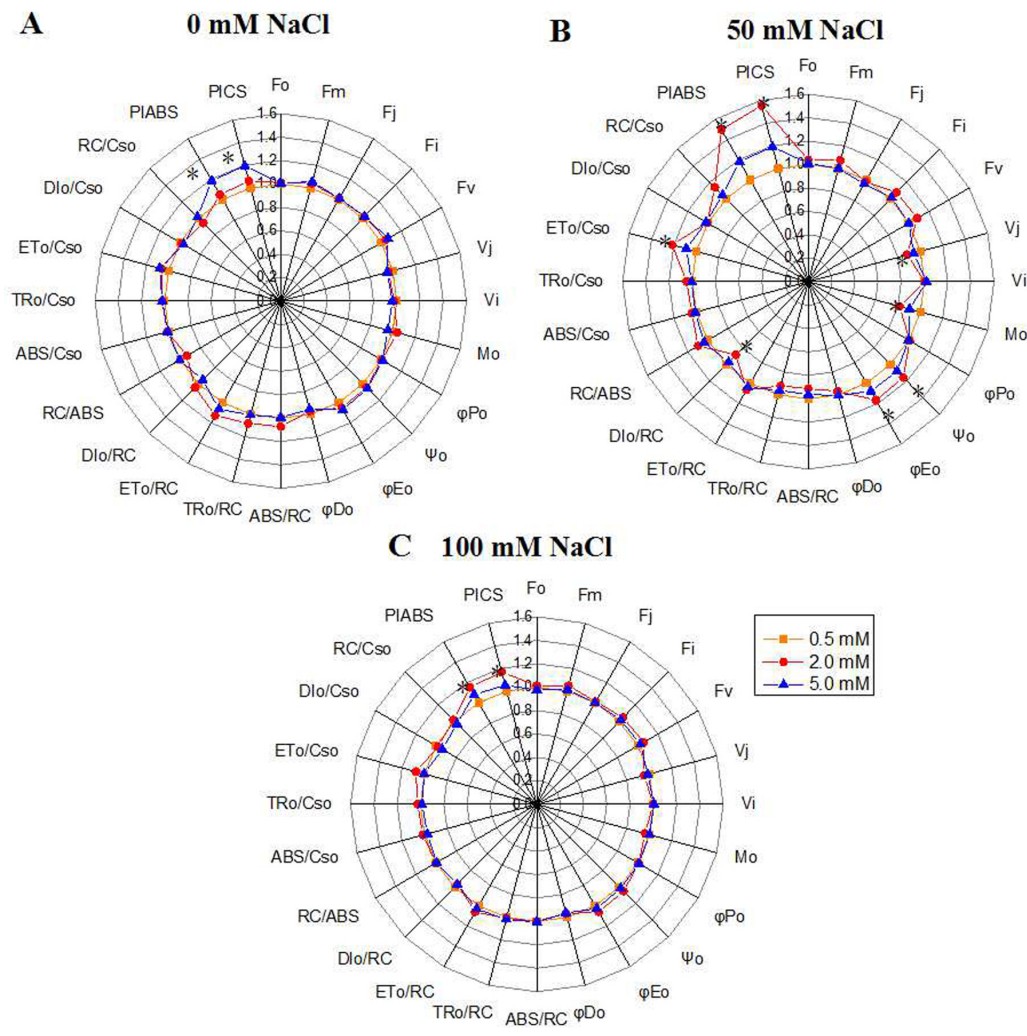

**Figure 3** "Radar plots" of picked parameters characterizing influence of nitrogen concentration (0.5, 2.0, 5.0 mM N) on PS II of annual ryegrass. The annual ryegrass leaves exposed to 0 mM (A), 50 mM (B), 100 mM (C) NaCl stress respectively. All values are shown as percent of control. The parameters of plants grown under 0.5 mM nitrogen concentration were set as control. Control = 1. * indicate significant differences of Chlorophyll fluorescence parameters between different N levels at $P < 0.05$ by Student's t–test under 0, 50, 100 mM NaCl respectively.

in the absence of salt stress, there was no significant difference of MDA content among the three N levels. When the plants grown under 2.0 mM N were exposed to a relatively lower NaCl treatment (50 mM), the MDA content declined significantly compared to the control. In addition, the MDA content of plants grown under higher N concentration were significantly lower compared to those grown under 0.5 mM N (Fig. 4A). The electrolyte leakage (EL) in the leaves of ryegrass increased with the increase of NaCl concentration under all three N levels. When exposed to 100 mM NaCl, The EL in the leaves of ryegrass grown under higher N concentration was significantly lower compared to that grown under 0.5 mM N. Moreover, under 100 mM NaCl stress, the EL of ryegrass grown under

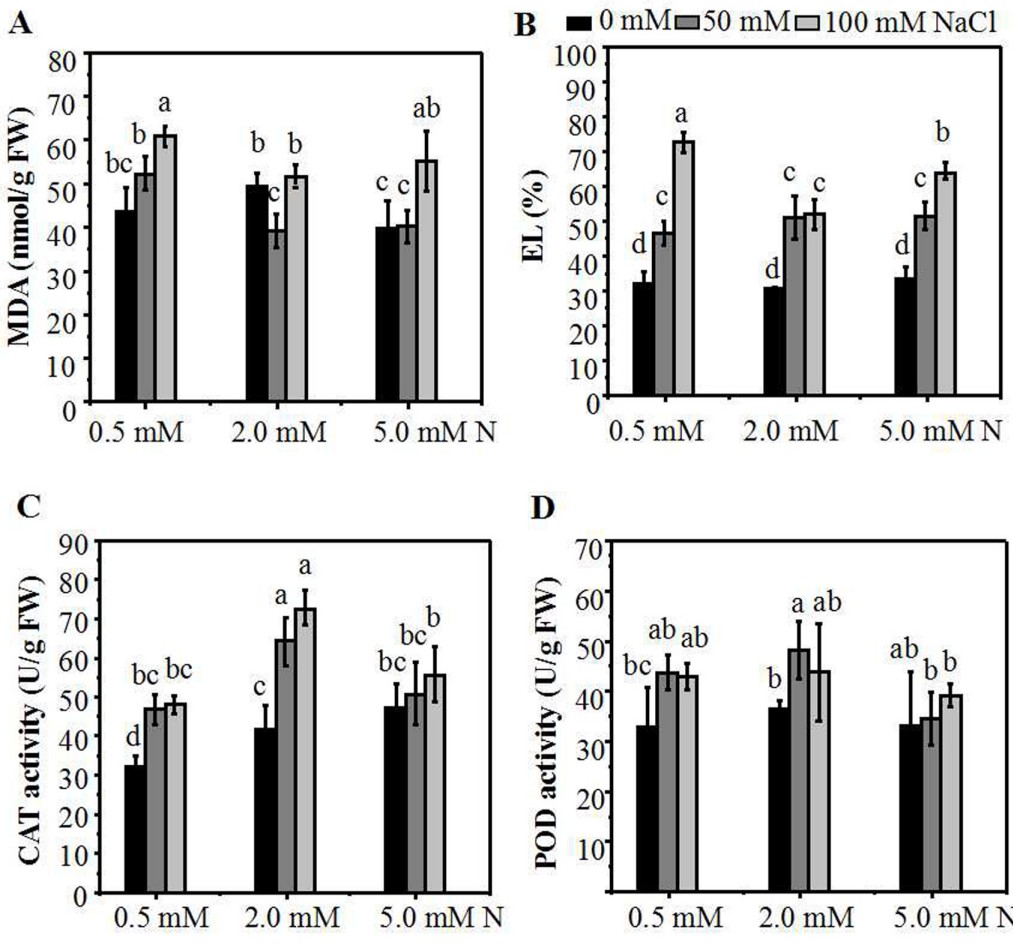

**Figure 4 The membrane damage degree and antioxidant enzymes activities of the annual ryegrass leaves.** MDA content (A), EL (B), catalase (CAT) (C) or peroxidase (POD) (D) activity in the leaves of annual ryegrass grown with different nitrogen concentrations (0.5, 2.0, 5.0 mM N) exposed to different NaCl level (0, 50, 100 mM NaCl) respectively. Different letters above the columns indicate statistically significant differences at P ¡ 0.05 by Student's t–test.

2.0 mM N was significantly lower than that grown under 5.0 mM N (Fig. 4B). The lipid peroxidation levels and activities of antioxidant enzymes of the leaves were also determined. With the increase of NaCl concentration, the CAT enzyme activity rose under all N levels. Under NaCl treatments, the CAT activity of ryegrass seedlings cultured at 2.0 mM N was the highest compared with that of plants cultured at 0.5 mM N or 5.0 mM N (Fig. 4C). The POD enzyme activity had no obvious regularity with the N levels. However, when exposed to 50 mM NaCl, the POD activity of seedlings grown under 2.0 mM N was higher than seedlings grown under 5.0 mM N (Fig. 4D). These results suggested that ryegrass cultured in 2.0 mM N solution might improve the activities of certain antioxidant enzymes and enhance the salt-tolerance ability of ryegrass, especially at relatively low NaCl level.
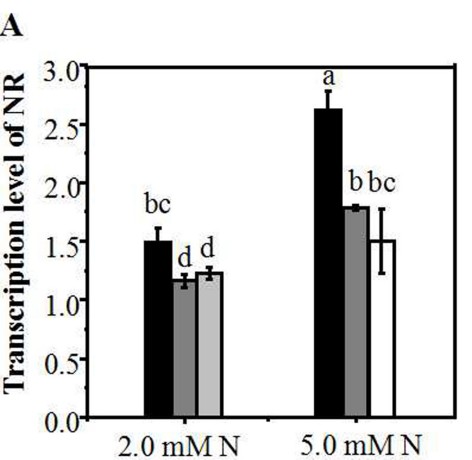
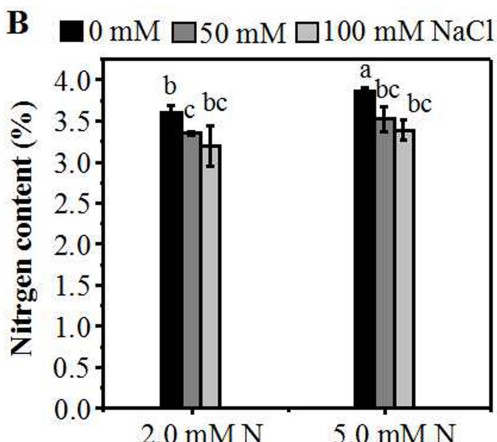

**Figure 5 Relative expression of N metabolism-related genes and nitrogen content of leaves grown under different conditions.** (A) NR expression in the leaves of annual ryegrass grown under different nitrogen concentration (2.0, 5.0 mM) exposed to different salt stress for 12 hours (0, 50, 100 mM NaCl) respectively; (B) Nitrogen content of leaves grown with different nitrogen concentrations exposed to different salt stress for 10 days respectively. Different letters above the columns indicate statistically significant differences at $P < 0.05$ by Student's $t$-test.

## Effect of different N treatment on the N content and N assimilation-related genes under NaCl stress

To investigate the influence of different N concentrations on N assimilation under NaCl stress, we assessed the expression level of *NR* gene in the leaves of ryegrass. Without NaCl treatment, the level of the *NR* expression in the leaves increased with the increase of N concentration. A reduction in N (2.0 mM) caused a significant decline in mRNA expression of *NR* (Fig. 5A), as compared with 5.0 mM N-applied plants. When plants were cultured with 5.0 mM N, the level of *NR* gene expression showed a significantly decrease with the increase of salt concentration. However, under 2.0 mM N, the suppression degree of *NR* expression by salt stress was relatively lower. Compared with 0 mM NaCl, the expression level of *NR* of 2.0 mM N-supplied plants showed no significant decrease when exposed to 100 mM NaCl (Fig. 5A). Under the combined treatment of nitrogen and salt, the homolog gene of *GS* showed a similar expression response pattern with *NR* (Fig. S2A). The expression of the *NRT* gene was induced when plants were exposed to a relatively low NaCl level (50 mM). However, there was no significant difference in the homolog of one *NRT* gene expression between plants grown with 5.0 mM N and 2.0 mM N under NaCl treatment (Fig. S2B). When exposed to NaCl, the nitrogen content of leaves grown under 2.0 mM N or 5.0 mM N showed a significant decline compared to the control condition, respectively. Without NaCl treatment, the N content of ryegrass leaves grown under 5.0 mM N was higher. However, under salt treatment, the N content of leaves showed no significant difference between 5.0 mM N and 2.0 mM N application (Fig. 5B).

## DISCUSSION

Plant response to salt stress is a complex phenomenon that involves both morphological, physiological, and biochemical processes. It has been reported that the application of N may alleviate salt stress-induced phytotoxicity (*Correia et al., 2005*; *Siddiqui et al., 2012*; *Singh, Singh & Prasad, 2016*). Salt stress effect on plants consists of $Na^+$ and $Chl^-$ ion toxicity, osmotic effects, and nutrient imbalance (*Kohler et al., 2009*; *Shannon, 1997*). Nitrogen fertilization plays a critical role on the growth and development of the many plants, and appropriate N could help to mitigate the damage caused by nutritional imbalances due to saline irrigation (*Al-Rawahy, Stroehlein & Pessarakli, 1992*; *Borzouei et al., 2014*; *Fan et al., 2013*; *Duan & Chang, 2017*). However, related researches showed that rather than N alone, plant growth was significantly affected by the interaction between soil salinity and N (*Papadopoulos & Rendig, 1983*; *Chen et al., 2010*). In this experiment, exogenous N application to annual ryegrass seedlings significantly increased the *in planta* N content, plant height and the biomass, but there was a concentration effect. In the absence of exogenous salt, the increment of plant height and biomass was directly proportional to the N level, and peaked at 5.0 mM N. However, after exposure to NaCl, the plant height and the relative increase of biomass peaked at the N level of 2.0 mM (Fig. 1). Moreover, ultra-low N seriously inhibited the growth of ryegrass under both control and salt conditions (Fig. S1). These results were consistent with results detected in cotton (*Chen et al., 2010*). Also a previous study on annual ryegrass reported that increasing N application could promote shoot growth under under salt concentrations of 2.0 dS/m (below 20 mM) and 11.2 dS/m (over 100 mM) (*Sagi et al., 1997*). However, we noticed that moderate reduction of N application had a maximum promotion effect on plant growth. The salt concentration used for our treatments are 50 mM and 100 mM, which are different from the salt concentration used in Sagi's experiments. This difference may be attributed to different culture conditions as well as levels of salinity. In this study, we focus mainly on the optimization of nitrogen application at lower salt concentration. Moreover, N content was also positively correlated with the amount of N applied which peaked at the N level of 5.0 mM without salt treatment. However, under salt treatments, there was no significant difference in N content of ryegrass leaves between 5.0 mM and 2.0 mM N application (Fig. 5B). External cues such as salt can stimulate the production of ROS which can further damage lipids in plant cells (*Kohler et al., 2009*). Accumulation of N-containing compounds has been reported to participate in salt response such as osmotic adjustment and ROS scavenging (*Dluzniewska et al., 2007*; *Ehlting et al., 2007*; *Homaee, Feddes & Dirksen, 2002*; *Mansour, 2000*; *Song et al., 2006*; *Sudmalis et al., 2018*). In this study, by reducing MDA content and elevating certain antioxidant enzyme activities, moderately low N application could reduce the damage to the membrane of ryegrass seedlings caused by salt stress, especially at relatively low NaCl treatment. Together, these results indicated that saline habitats might change the N requirement of ryegrass seedlings. Excessive or ultra-low N applications both have counter effects on the growth or salt resistance of annual ryegrass under low level of salt stress.

Under salt stress, chlorophyll *a* fluorescence transient is a useful parameter for reflecting the primary reaction alternations of PSII, which is more sensitive than photosystem I (PS I). To investigate PSII behaviors in O-J-I-P transient, JIP test is always used to quantify the derived photochemical parameters (*Apostolova et al., 2006*; *Sayed, 2003*; *Stirbet et al., 2014*). In this study, after exposure to NaCl, the nitrogen application level had a significant effect on fluorescent transients, especially the J and P steps (Fig. 2). N deficiency and N over application under salt stress might lead to the photosynthetic electron transport traffic jam, especially beyond $Q_A^-$ (Figs. 2B, 2C). With the increase of N level, the $PI_{ABS}$, which could accurately reflect the state of plant photosynthetic apparatus, increased without NaCl treatment (Fig. 2A), indicating that N could promote the primary photochemical reactions of PSII in the waterside. However, under salt stress, excessive or deficient N application slowed down the promotion of the primary photochemical reaction of the PSII (Fig. 2B, Fig. 2C). In addition, the accumulated amount of $Q_A^-$ ($V_j$) and the relative speed of $Q_A$ deoxidation ($M_0$) (*Strasser, 1997*; *Strasserf & Srivastava, 1995*; *Force, Critchley & Rensen, 2003*) of plants grown under moderately low N were lower than those grown under other N conditions, indicating that leaves grown under moderate N level have a higher electron transport rate between $Q_A$ and $Q_B$, thus reducing the accumulation amount of $Q_A^-$ andincreasing the photochemical reaction efficiency (*Allakhverdiev & Murata, 2004*. The increase of $\Psi_0$ and $\Phi E_0$ of plants grown under 2.0 mM N indicated that leaves use more energy for photochemical reaction and electron-transfer process, thus producing more NADPH for carbon assimilation and confirming that leaves have the optimal energy distribution under certain salt level (*Strasser, Tsimilli-Michael & Srivastava, 2004*). The leaves of annual ryegrass grown under 2.0 mM N also had a greater reaction center density $RC/CS_0$ and higher $ET_0/CS_0$ but lower $DI_0/CS_0$ than those grown under other N conditions. This pattern indicated that the specific activity of a unit cross-sectional area of leaves grown under moderately low N was stronger than that grown under other N conditions, reducing the energy burden of a unit reaction center. We also noticed that, under the lowest NaCl treatment (50 mM), the application of 2.0 mM N had the best effect on alleviating salt stress. Under 50 mM NaCl treatment, the physiological indexes of annual ryegrass seedlings seemed to be less affected, and therefore the seedlings might be more sensitive to the promotion of nitrogen application. Thus, we proposed that the optimum amount of N might promote primary photochemical reactions of PSII under certain level of NaCl treatment.

After absorption as ammonia, N can be directly assimilated by plants. While after absorption as nitrate, N must first be reduced by nitrate reductase and sub-acid reductase enzymes. NR can reduce nitrate N to ammonium, which has also has important effects on photosynthesis and other processes related to N metabolism (*Xu, Fan & Miller, 2012*). Reports have shown that $NO_3^-$ has a significant effect on the induction of NR expression. From this experiment, under control condition, the level of *NR* gene expression in leaves increased with the increase of N concentration (Figs. 5A, 5B), which is consistent with the previous reports (*Oaks, 1993*). However, when the seedlings were treated with NaCl, the *NR* expression level was significantly declined at a higher N level (5.0 mM). On the contrary, at a moderately low N level (2.0 mM), the *NR* expression level is relatively

low without NaCl treatment, but the degree of reduction is moderate when exposed to NaCl. The *GS* gene expression displayed a similar trend with the *NR* gene under the interaction between salt and nitrogen conditions, indicating an interactive response mechanism between N assimilation-related genes (Fig. S2A). Therefore, it can be observed that moderate N application might help annual ryegrass maintain the expression level of N assimilation-related gene (Fig. 5A) and further maintain the nitrogen content under salt stress (Fig. 5B). However, when excessive N was applied under salt stress, the *NR* expression was significantly increased, indicating that N assimilation was strengthened; it might then compete with photosynthetic carbon for ATP and NADPH and increase the burden of photosynthetic electron transfer. The competition result might lead to a decline in the overall activity of PSII of annual ryegrass (PI$_{ABS}$) (Fig. 2). Under nitrogen deficiency, the reduced absorption of nitrogen might reduce the consumption of nitrogen assimilation reducing power, most of which are derived from photosynthesis, thus resulting in the accumulation of chloroplast NADPH. The over-accumulation of NADPH could inhibit the photosynthetic efficiency and cause excessive production of ROS (Fig. 3A), leading to increased cell membrane damage, which may in turn lead to reduced photosynthetic efficiency (Figs. 2B, 2C, 2D). Nitrogen is also a constituent of chlorophyll which is not only the most important pigment molecules of photosynthesis involved in energy absorption and transmission but also the essential electron mediator during electron transport. Studies showed that the nitrogen content of leaves is constant with the photosynthetic capacity (*Grassi et al., 2005*; *Kattge et al., 2009*). Through this experiment, we can deduce that under different salt stress condition, the appropriate addition of N can increase relative chlorophyll content of plants. However, the relative chlorophyll content is only positively correlated with N levels within a certain range (0.5–2.0 mM) and should be reduced beyond a certain range (Fig. 2D). The moderately supply of N under salt stress increased the content of chlorophyll and might increase the light-harvesting ability, partly contributing to the up-regulated photosynthetic performance index. Based on the above studies, it can be seen that moderately low N application under low level of salt stress might help annual ryegrass maintain the expression level of N assimilation-related gene and then maintain the leaf N content of the plant, which might in turn cause changes in chlorophyll content, further avoiding the negative effect on photosynthetic capacity.

## CONCLUSION

To investigate the possible mechanism of moderately low nitrogen-mediated alleviation of NaCl stress, the degree of lipid peroxidation, antioxidant enzyme activity alternation, changes of photosynthesis performance and nitrogen assimilation were analyzed in this study. In summary, under low salt stress, the demand for N may have declined, while moderately reducing N application could help to alleviate the salt-induced damage in annual ryegrass mainly by alleviating the damage caused by ROS and promoting the performance of photosynthesis and nitrogen metabolism. Further, in order to enhance plant growth and increase nitrogen use efficiency, the optimum application of nitrogen fertilizer needs to be controlled to match the plant needs at each growth stage and to adapt to different saline environment.

## ACKNOWLEDGEMENTS

We thank Dr. Yongzhe Ren (Henan Agriculture University) for the valuable advice on the design of the experiments.

### Funding

This work was supported by the China National Natural Science Foundations (grant no. 31801892) and the Natural Science Foundation of Shandong Province, China (grant no. ZR2019PC012). The funders had no role in study design, data collection and analysis, decision to publish, or preparation of the manuscript.

### Grant Disclosures

The following grant information was disclosed by the authors:
China National Natural Science Foundations: 31801892.
Natural Science Foundation of Shandong Province, China: ZR2019PC012.

### Competing Interests

The authors declare there are no competing interests.

### Author Contributions

- An Shao conceived and designed the experiments, analyzed the data, prepared figures and/or tables, authored or reviewed drafts of the paper, and approved the final draft.
- Zhichao Sun conceived and designed the experiments, performed the experiments, prepared figures and/or tables, authored or reviewed drafts of the paper, and approved the final draft.
- Shugao Fan and Xiao Xu conceived and designed the experiments, performed the experiments, prepared figures and/or tables, and approved the final draft.
- Wei Wang conceived and designed the experiments, analyzed the data, prepared figures and/or tables, and approved the final draft.
- Erick Amombo performed the experiments, prepared figures and/or tables, authored or reviewed drafts of the paper, and approved the final draft.
- Yanling Yin analyzed the data, prepared figures and/or tables, authored or reviewed drafts of the paper, and approved the final draft.
- Xiaoning Li, Guangyang Wang and Jinmin Fu analyzed the data, authored or reviewed drafts of the paper, and approved the final draft.
- Hongli Wang conceived and designed the experiments, authored or reviewed drafts of the paper, and approved the final draft.

### Data Availability

The raw data are available in the Supplemental Files.

## Supplemental Information

Supplemental information for this article can be found online at http://dx.doi.org/10.7717/peerj.10427#supplemental-information.

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
