# Peer review of "Moderately low nitrogen application mitigate the negative effects of salt stress on annual ryegrass seedlings"

_PeerJ, doi:10.7717/peerj.10427_

## Round 0.1 · original submission · Minor Revisions

Based on the reviews from two anonymous reviewers, before being accepted, your manuscript requires some modifications. Particularly, grammar corrections and language polishing are required to further improve the quality of this paper.

Reviewer 1 ·

Basic reporting

The language is readable, but there are still many errors. I suggest that the manuscript should be extensively modified by a native English speaker.
The article includes sufficient introduction and background with relative references.
I did not see Figure legends of Fig. S1 and S2.
I cannot find Table S1, which should show the primers.
Raw data are provided.
The sections are right.

Experimental design

N application can alleviate the salt stress on plants. The authors aim to reveal how N concentrations affect the growth and photochemical parameters of feeding annual ryegrass under low salt stress. That is an important question in the environment science field.
The authors systematically analyzed the growth and photochemical parameters of feeding annual ryegrass at different treatments. The methodology is reliable and clearly described.
However, one problem is that the authors did not properly organize the data. In some figures, the x axis is N concentrations, while in others the x axis is salt concentrations.

Validity of the findings

Repeats were properly performed.
The data were statistically analyzed.
The conclusion is related to the main question asked by the research and supported by the data.

Additional comments

The following are specific comments and suggestions. The authors should be aware that the errors are much more than listed below. Therefore, the manuscript has to be carefully and extensively modified.
1. Line 18: damage on plants
2. Line 23: full name of “OJIP”
3. Line 42: remove “studies have shown that”
4. Line 47: reduce its damage; change evolvement to evolution
5. Line 53: change regulate to improve
6. Line 54: rephrase the sentence “The inorganic N …”
7. The authors analyzed the effects of N concentrations on plant growth. They show the data of 2, 5, 10 mM in Fig.1 but the data of 0.5, 2, 5 mM in Fig. S1. Why not put these data in one figure?
8. Please mark the x axis in Fig. S1D.
9. Line 264: remove “the”
10. Line 277-279: The authors compared their results to the findings of Sagi et al., 1997. However, they should convert salinity to NaCl concentration for comparison.
11. Line 300: The results are not from Fig. 1
12. Figure 3 legend: Rephrase the sentence “*indicate parameters statistically significant between different N levels under the same NaCl level.”
13. Line 326-327: what it represents?
14. Line 332: is consistent with
15. Acknowledgement: The authors should mention funding sources and people who helped them prepare the manuscript, instead of reviewers.

·

Basic reporting

This is a very meaningful manuscript with professional language, structure, and figures.

Experimental design

Rigorous experimental design.

Validity of the findings

The results are solid and valid.

Additional comments

I suggest adding “seedings” to the topic since the research object is the seedlings rather than the whole growth period.

To update some new literatures beyond the classic literatures to support the research results.

---

## Round 0.2 · accepted · Accept

The reviewers' comments have been addressed, and thus your manuscript can be accepted. Thank you for your contribution to the PeerJ. Waiting for your next work.